# Assessing the relative contributions of mosaic and regulatory developmental modes from single-cell trajectories

**Solène Song**⬤*, **Paul Villoutreix**⬤*

Aix-Marseille Université, MMG, INSERM U1251, Turing Centre for Living Systems, Marseille, France

* solene.song@univ-amu.fr (SS); paul.villoutreix@univ-amu.fr (PV)

## Abstract

Development is a complex process driven by coordinated cell proliferation, differentiation, and spatial organization. Classically, two ways to specify cell types during development have been hypothesized: the mosaic and regulative modes. In the mosaic mode, a particular cell isolated from the rest of the embryo will still give rise to progeny with the same fate as expected in normal development, relying on lineage-inherited factors. In contrast, in the regulative mode, the fate of a cell depends on its interactions with its environment and thus relies on space-dependent factors. While both modes often co-exist, their relative contributions remain poorly quantified at single-cell resolution. We present a novel approach to measure these contributions from single-cell data from *C. elegans* development. The invariant lineage of *C. elegans* allows the integration of spatial positions, lineage relationships, and protein expression data. Using single-cell protein expression profiles as a readout of cell state, we define two quantifiable metrics: 1) a proxy for the contribution of the mosaic mode, computed as the strength of the relationship between the cell-cell lineage distance and the cell-cell expression distance, 2) a proxy for the contribution of the regulative mode, computed as the strength of the relationship between the cell-cell context distance - capturing spatial neighborhood similarity - and the cell-cell expression distance. To validate these metrics, we compared empirical results from *C. elegans* to artificial models with defined developmental rules. Our analysis reveals the coexistence of mosaic and regulative modes, with their relative contributions varying across tissues and developmental stages. For example, in skin tissue, the mosaic mode dominates in early development, while the regulative mode prevails later. Our approach offers a quantitative, unbiased, and perturbation-free method to study fundamental principles of developmental biology.

the terms of the Creative Commons Attribution License, which permits unrestricted use, distribution, and reproduction in any medium, provided the original author and source are credited.

**Data availability statement:** Our work uses data published in (i) Ma, Xuehua, et al. "A 4D single-cell protein atlas of transcription factors delineates spatiotemporal patterning during embryogenesis." Nature Methods 18.8 (2021): 893-902.-> link to the original dataset: https://static-content.springer.com/esm/art%3A10.1038%2Fs41592-021-01216-1/MediaObjects/41592_2021_1216_MOESM5_ESM.xlsx and (ii) Li, Xiaoyu, et al. "Systems properties and spatiotemporal regulation of cell position variability during embryogenesis." Cell reports 26.2 (2019): 313-321.-> link to the original dataset: https://usegalaxy.org/published/history?id=b5b10395dc3f1cb2. The code is available on the following GitHub repository: https://github.com/VILLOUTREIXLab/mosaic_vs_regulative.

**Funding:** SS was an employee of Aix-Marseille University and funded by the "Investissements d'Avenir" French Government program managed by the French National Research Agency (ANR-16-CONV-0001) through the Excellence Initiative of Aix-Marseille University - A*MIDEX. PV was an employee of INSERM. The funders had no role in study design, data collection and analysis, decision to publish, or preparation of the manuscript.

## Author summary

Understanding how a single cell is transformed into an organism composed of a multitude of cells with precise functions and positions requires understanding how cells choose their fates. Two main mechanisms are usually assumed: the mosaic mode and the regulative mode. In the mosaic mode, the fate of a cell is predetermined by inherited factors. Meanwhile, in the regulative mode, the fate of a cell depends on signals from its surroundings. We used the nematode *C. elegans*, a model organism with a well-mapped development, to propose a way to assess the respective contributions of these modes. By analyzing individual cells, their positions, lineage relationships, and protein patterns, we developed two measures to quantify these contributions. One measure shows how much cell fate depends on inheritance (mosaic), while the other captures the impact of neighboring cells (regulative). Our findings show that both modes coexist, but their importance shifts over time and between tissues. For example, early skin development is more mosaic, while later stages rely on regulative processes. This method gives us a new and precise way to study how organisms build themselves without disrupting the process through experimental perturbations.

## Introduction

Understanding how a single fertilized egg (zygote) gives rise to a fully functional organism is a central question in developmental biology. Cells must acquire the correct fate at the correct location, progressing through successive intermediate states commonly measured by transcriptomic profiles. Classically, two modes of development are hypothesized: mosaic and regulative [1] (Fig 1). In mosaic development, a cell isolated from the rest of the embryo will still produce the same progeny as in normal development [2] whereas in regulative development, the fate of a cell depends on its interactions with its environment. Both modes often co-exist [1].

Experimental embryology has historically assessed these contributions using perturbative approaches, such as cell isolation, grafting, or embryo bisection. Roux (1888) proposed an experiment where one blastomere in the two-cell frog embryo was destroyed, yielding a half-embryo, illustrating mosaic development. In contrast, Driesch (1902) observed that dissociated sea urchin blastomeres sometimes produced smaller but complete embryos, demonstrating regulative capacity. Such emblematic experiments, although impressive, are limited to specific stages, become intractable as cell numbers increase, and are highly disruptive. Microscopy imaging, a less disruptive technique, suggests that both mosaic and regulative modes are involved [1]. However, there is no systematic method to quantitatively assess the relative contributions of these modes throughout the entire course of development. Here, we present a method paper addressing this gap and illustrate our approach by applying it to *C. elegans* development.

Importantly, *C. elegans* displays invariant lineage, meaning that patterns of division and differentiation are reproducible across individuals. Lineage invariance has intuitively led to the assumption of mosaic development [1]: since the lineage is so

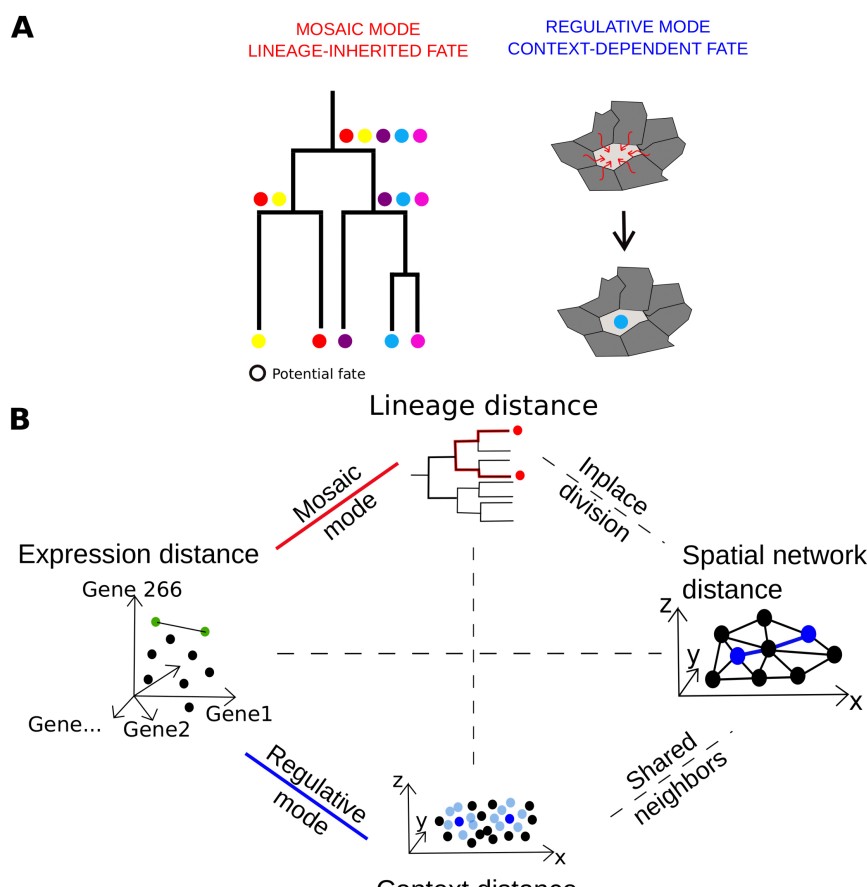

**Fig 1**. **Conceptual overview of developmental modes and cell-pair distance measures A) Schematic illustration of the distinction between mosaic mode and regulative mode.** In the mosaic mode, the colored dots represent potential cell fates that a cell can take, transmitted to the daughters at each division. In the regulative mode, the cell fate of the central cell is determined by the states of its neighbors **B)** Schematic illustration of the four distances between cell pairs, and the relations between them. The *lineage distance* is the length of the shortest path between two cells in the lineage tree taken as a graph. The *spatial network distance* is the length of the shortest path in the Delaunay graph connecting the cells spatial positions. The *expression distance* is the correlation distance between two expression profiles. The *context distance* between cell A and cell B is the correlation distance between the mean expression profile of the neighbors of A and the mean expression profile of the neighbors of B.

robustly conserved, the information for differentiation must be passed on from mother to daughter cell. However, invariant lineage also entails reproducibility in cell positioning and neighborhood, so fate could equally result from reproducible cell-cell spatial interactions. The role of cell-cell spatial interactions in *C. elegans* development has been highlighted before [3,4], tested experimentally by laser ablation [5], and shown to be required for known signaling pathways such as early Notch signaling [6], and late Notch signaling [7,8]. Overall, both modes contribute to the development of *C. elegans* and their respective contributions have not yet been quantified.

A unique feature of *C. elegans*, made possible by its invariant lineage, is the possibility to combine spatial positions, lineage relationships and protein expression data. Each cell produced during development has a defined identity, and a corresponding counterpart across embryos. In this study, we integrate two published datasets: spatial positions and lineage relationships obtained from nuclear tracking [3,9] and protein expression profiles derived from imaging-based measurements of 266 transcription factors [10].

Simultaneous measurement of these features in other organisms remains highly challenging. Current technologies allow combining expression and spatial position, such as spatial transcriptomics [11], or combining lineage and expression through lineage tracing coupled with scRNA-seq [12]. When complete information on the three aspects is lacking, inference methods can sometimes bridge the gap, for example, for predicting spatial positions from expression patterns, but at the cost of simplifying assumptions [13,14]. Emerging techniques like intMEMOIR [15] hold promise for integrated measurements, but remain technically demanding. Therefore, our integrated *C. elegans* dataset provides a uniquely valuable ground truth and an ideal model system for studying the interplay between lineage, spatial organization, and expression dynamics, until experimental and computational approaches mature to produce comparable datasets in other systems.

Our method quantifies the relative contributions of mosaic and regulative modes. For this purpose, it relies on comparing different measures of distance between pairs of cells: *lineage distance*, *expression distance*, and *context distance*, where context reflects the expression profiles of the respective neighbors of the two cells. The central assumption is that, under a predominantly mosaic mode, cells that are closely related in the lineage tree are expected to exhibit higher similarity in their expression profiles. Conversely, under a predominantly regulative mode, cells with similar contexts, regardless of their lineage relatedness, should display greater similarity in expression profiles. The use of the correlation between *lineage distance* and *expression distance* as a proxy for mosaic mode rests on a simplifying assumption: during each cell division, most molecular information from the mother cell is transmitted to both daughter cells. As a result, their expression profiles remain close to that of the mother and, consequently, more similar to each other than to those of more distantly related cells. The differences observed between sister cells are expected to include a smaller asymmetric component, either inherited or acquired during their life cycles, that drives their gradual divergence.

There are specific situations in which this assumption does not seem to apply, at least for a part of the inherited information. For example, during the early divisions of *C. elegans*, the asymmetric segregation of P-granules specifies the germ line [16–18], leading to an abrupt fate asymmetry. Lineage-relatedness and expression similarity also clearly diverge for left-right symmetric cells, which can be distantly related in the lineage yet exhibit highly similar expression profiles. The latter case may reflect either regulative processes or lineage-dependent inheritance mechanisms that fall outside the scope of our simplifying assumption. Nevertheless, our assumption remains practical, simple, and hopefully broadly applicable, since we expect it to hold for most cell pairs during development, and particularly when large sets of gene expression profiles are considered.

We developed quantitative measures that serve as proxies for how much information is passed on through lineage-inherited factors versus interactions with neighbors, which we refer to as the contribution of the mosaic mode $W_{mosaic}$ and the contribution of the regulative mode $W_{regulative}$, respectively. More formally, as detailed in the Materials and Methods section, $W_{mosaic}$ is built from stratified correlations [19] between *lineage distance* and *expression distance* for all cell pairs at all time points. For a given time point, the distribution of these stratified correlations is compared with a random expression model, and $W_{mosaic}$ represents the difference between the real *C. elegans* data and the random expression model. $W_{regulative}$ is defined analogously, but replacing *lineage distance* with *context distance*. These two metrics, while they do not imply direct causality, can suggest the presence of underlying biological mechanisms [20].

By analyzing separately the subpopulations of cells that give rise to each of the five labelled distinct tissues in *C. elegans* [10], we show that the mosaic mode contribution is significant in all tissues while the regulative mode contribution depends on time and on the tissue: it is significant earlier in neuron and skin than in muscle and pharynx, whereas it never is significant in intestine development during the period of observation. We then refine this result by highlighting a population of cells in which the measured regulative mode seems dominant. Finally we discuss how both differentiation and cell rearrangement contribute to the measured regulative mode signal.

## Materials and methods

We introduce a method to quantify the respective contributions of mosaic and regulative modes in *C. elegans* development, denoted $W_{mosaic}$ and $W_{regulative}$. An intuitive explanation of these measures is provided in the *Introduction* and their detailed construction is described in the subsection *Calculations of the contributions $W_{mosaic}$ and $W_{regulative}$*, following the presentation of all required inputs.

This method can be applied only when lineage relationships, spatial positions, and temporal expression profiles of cells are available, as is the case for *C. elegans* (see *Datasets* subsection). Using this information, we define four distance measures: (1) *lineage distance*, (2) *spatial network distance*, (3) *expression distance* and (4) *context distance*. The precise definitions of these distances are provided in the *Distances* subsection.

The construction of $W_{mosaic}$ and $W_{regulative}$ requires a random null model used to generate reference statistical distributions (described below). To validate our metrics on a simple and extreme case, we designed a model that represents purely lineage-defined expression profiles, allowing us to test whether under this model $W_{mosaic}$ is high and $W_{regulative}$ is low as expected. The random null model and lineage-defined model are described in detail in the *Artificial models* subsection.

In addition, we introduce the definition of *octants*, which allows us to identify pairs of cells that are expected to undergo exclusive regulative development, disentangled from mosaic influences, namely those that share similar expression profiles and similar neighbors' expression profiles, yet are distant with respect to the *lineage distance* and the *spatial network distance*.

Finally, we introduce how to compute the rearrangement rate, a process that affects the spatial organization of cells and consequently the contribution of the regulative mode $W_{regulative}$.

## Datasets

For most cells in *C. elegans* development, the following information is available:

- their position in the lineage tree
- their position in the three-dimensional physical space
- their partial protein expression profile

We combine two published datasets to obtain spatial and expression information:

- cell positions in 3D space, obtained from one embryo among 28 of the datasets of Li et al. [3], and
- protein expression profiles from Ma et al. [10], who combined reporter-line measurements from hundreds of embryos to reconstruct expression dynamics for all cells throughout the lineage. The expression profiles are partial because the authors restricted their measures to a curated subset of transcription factors likely to be relevant for development.

The invariant *C. elegans* lineage was first described in [5]. It is co-measured, together with the feature of interest, in all datasets, enabling their integration.

**Spatial positions.** The dataset containing the spatial positions was originally published in [3], where the 3D positions were measured by tracking nuclei with ubiquitously expressed mCherry. The time points range from t=0 to t=190, with 75-second intervals. Out of the 28 embryos recorded in their dataset, we chose the embryo number 13, which had the highest number of tracked cells. At the final tracked time point, t=190 (approximately 4 hours), the embryo has completed all but its last round of cell division, resulting in 380 cells present at that moment. Cumulatively, from t=0 to t=190, the embryo has generated a total of 762 cells. This implies that the lineage tree, as it stands at t=190, consists of 762 cell IDs (including 380 leaves and 382 ancestor cells) out of a total 1341 cells produced across the entire development.

**Expression profiles.** The protein expression dataset was originally published in [10]. The dataset was obtained by fluorescent microscopy imaging of the protein expression levels of 266 transcription factors using protein-fusion reporters. Thanks to *C. elegans* lineage invariance, the expression profiles are unambiguously paired with the cell identities. We have the expressions of all cells including the last division, except for the first cells P0, P1 and AB, with a total of 1204 cells tracked out of a total of 1341 cells. The terminal cells come with their tissue annotation: pharynx, skin, neuron, muscle and intestine. We propagate this terminal tissue label backward following daughter to mother branches to identify all the cells that contribute to a given tissue. The contributions of the mosaic and regulative modes are calculated for these groups of cells separately.

## Distances

The computations for the contributions $W_{mosaic}$ and $W_{regulative}$ rely on the correlation between the *expression distances* and, respectively, *lineage distances* and *context distances* between pairs of cells. We detail here the definitions of these distances.

**The *lineage distance*.** measures how closely two cells are related. For example, if two cells are sisters, this distance should be very low, and if two cells come from different parts of the lineage, this distance should be high. To measure this quantitatively, we first encode the lineage tree as a tree-graph where the nodes are the cells and the undirected edges encode the ascendant/descendant relationships. We then compute the length of the shortest path between two cells in the lineage tree as the number of edges in the shortest path from one cell to the other in the tree-graph. $d_{lineage}(A, B) = length(ShortestPath_{lineagetree})(A, B)$.

**The *expression distance*.** measures how similar two expression profiles are. To measure this distance, we chose to use the correlation distance, well-adapted for expression profiles in a high-dimensional space (in contrast to euclidean distances) [21], defined as follows: $d_{expression}(A, B) = 1 - \frac{(\mathbf{x_A} - \bar{\mathbf{x_A}}) \cdot (\mathbf{x_B} - \bar{\mathbf{x_B}})}{||\mathbf{x_A} - \bar{\mathbf{x_A}}||_2 ||\mathbf{x_B} - \bar{\mathbf{x_B}}||_2}$ where $\mathbf{x_A}$ and $\mathbf{x_B}$ are vectors of dimension M=266 (number of genes measured), representing the expression profiles of cells A and B, and $\bar{\mathbf{x_A}} = \frac{1}{M} \sum_{i=1}^{M} x_{A,i}$ and $\bar{\mathbf{x_B}} = \frac{1}{M} \sum_{i=1}^{M} x_{B,i}$ the respective mean values of the elements of $x_A$ and $x_B$.

**The *spatial network distance*.** measures how close in the embryo two cells are. To circumvent potential biases related to the specific 3D coordinates of the cells in physical space, we chose to measure the distance as the length of the shortest path in the spatial network of cell connectivity. Because cell membrane information is not available, we approximate cell–cell connectivity by constructing a Delaunay graph in which each vertex is placed at the spatial position of an individual cell and the cell spatial positions correspond to the center of the detected nuclei from microscopy imaging [3]. This definition of *spatial network distance* is relevant with respect to cell-cell communication. The Delaunay graph is a good proxy for the network of cellular contacts, and under the assumption that most of the cell–cell communication happens through membrane contacts [22], the number of edges in the Delaunay graph between two cells gives the information of the relative distance between them. The *spatial network distance* is computed as $d_{SpatialNetwork}(A, B) = length(ShortestPath_{SpatialNetwork})(A, B)$.

**The *context distance*.** between two cells, A and B, is defined as the correlation distance between the mean expression profile of the direct neighbors of A and the mean expression profile of the direct neighbors of B. The neighbors $\mathcal{N}_A$ and $\mathcal{N}_B$ are the sets of cells which are one edge away in the spatial network respectively from *A* and *B*. The *context distance* is computed as : $d_{context}(A, B) = 1 - \frac{(\mu_A - \bar{\mu_A}) \cdot (\mu_B - \bar{\mu_B})}{||\mu_A - \bar{\mu_A}||_2 ||\mu_B - \bar{\mu_B}||_2}$ where $\mu_A = \frac{1}{|\mathcal{N}_A|} \sum_{n_i \in \mathcal{N}_A} \mathbf{x_{n_i}}$ is the mean expression profile of neighbors of A, $\mu_B = \frac{1}{|\mathcal{N}_B|} \sum_{n_i \in \mathcal{N}_B} \mathbf{x_{n_i}}$ is the mean expression profile of the neighbors of B, and $\bar{\mu_A}$ and $\bar{\mu_B}$ are the mean of the elements of $\mu_A$ and $\mu_B$.

Please note that when the number of cells is too small, most cell pairs share a large proportion of their neighbors. In such cases, a low *context distance* may not indicate that two cells have different sets of neighboring cells with similar expression profiles, but rather they share the same neighbors because they are located in the same spatial region. As

shown in S2 Fig, once the *spatial network distance* between cell pairs exceeds 3, they no longer share any neighbors. Additionally, we observed that the average ratio of shared neighbors between all cell pairs drops to less than 30% from 62 minutes onward.

## Artificial models

We built two artificial expression models as references to compare with the *C. elegans* statistics. The first one, the random null model, serves as a null model whose statistics are used in the calculation of the contributions $W_{mosaic}$ and $W_{regulative}$. The second one, the lineage-defined model, was built to be an extreme case of mosaic development.

**Random null model.** The expression profiles of cells were randomly shuffled across the lineage tree. Because the spatial positions were stored using the identity in the lineage tree, the expression profiles are also randomly distributed across space.

**Lineage-defined model.** Artificial expression vectors were generated to model an extreme case of lineage-dependent development. In this model, gene expression is transmitted identically from the mother to both daughter cells. During a cell cycle, expression changes by cumulative variations, diverging from its initial value. Because these changes persist in the same direction, cells progressively diverge in their expression profiles. We expect in this case that, given two cells, their *expression distance* correlates well with their *lineage distance*.

We first draw a random expression profile $x(0)$ for the egg P0 from a log-transformed uniform distribution from 0 to 1.5. Then, for each gene, the expression value across the lineage tree is updated as follows. A cell cycle is decomposed into $n_{steps} = 3$ steps. At each step $i$, the expression vector is updated as follows: $x(i + \Delta_i) = x(i) + v(i)\Delta_i$, where $\Delta_i$ is defined as a unit of time ($\Delta_i = 1$) and with $v(i) = v(i-1) + \epsilon(i)$, where $\epsilon(i)$ is drawn from a normal distribution centered at 0 with standard deviation $2/n_{steps}$. The initial speed $v(0)$ is drawn from a normal distribution centered in 0 and standard deviation 0.2. At each division, the first step of expression profile of the daughter cells are set to coincide with the last expression profile of the mother cell. Then each daughter cell's expression profile diverges from the mother following the updating rule above. The resulting unique expression profile per cell cycle that will be used for testing our metrics is obtained as the average of these three intermediate values of a cell cycle. When considering the mean expression over a cell cycle, these two descriptions are equivalent: cells that begin with the mother's expression profile and gradually diverge during the cycle can be viewed as inheriting expression largely symmetrically, with a smaller asymmetric component driving gradual divergence.

This process is inspired by the PROSSTT algorithm [23], which simulates single-cell RNA sequencing data as weighted mixtures of a small number of programs represented by vectors associated with each time point along a lineage tree. Our update procedure for simulating gene expression follows a similar logic to that of PROSSTT for generating the programs. We do not aim to produce realistic gene expression profiles but rather a simple and extreme behavior of progressive expression divergence.

The updating rule can also be interpreted as a random acceleration process, in which the expression profile represents the position of a walker whose speed follows Brownian motion, with its magnitude gradually increasing from the initial value.

200 gene expressions profiles were generated. As expected, the *expression distance* is highly correlated with the *lineage distance* ($r > 0.85$).

## Calculation of the contributions $W_{mosaic}$ and $W_{regulative}$

An intuitive approach would simply be to calculate the Pearson correlation between *expression distances* and *lineage distances* as a proxy for the contribution of mosaic mode, and the Pearson correlation between *expression distances* and *context distances* as a proxy for the contribution of regulative mode. However, these values are both mildly high, respectively 0.49 and 0.59 (mixing all time points), not indicating a strong relationship (S1 Fig). In addition, one can

immediately see as shown in Figs 1B and S1, that these relationships are intertwined through confounding effects. For example, because cells which just divided are still close to each other, it is expected that two cells close in lineage, which just branched off from a common ancestor are still close in space [24]. Because they are close in space, they would share a number of common neighbors, resulting in a small *context distance* too. To disentangle the relationships between these variables, we need to perform stratified correlations [19]. This means that to assess the contribution of mosaic mode, we compute the correlation between the *lineage distance* and the *expression distance* among pairs that have the same *context distance*. The same goes for regulative mode, for which we compute the correlation between the *context distance* and the *expression distance* among pairs that have the same *lineage distance*. The stratified correlations obtained this way must then be compared to a reference distribution to characterize the statistical significance of the values of $W_{mosaic}$ and $W_{regulative}$. This reference distribution comes from the random null model introduced previously.

**Calculation of the contribution of the mosaic mode $W_{mosaic}$.** The contribution of the mosaic mode $W_{mosaic}$ is computed for each time point. We use the Pearson correlation coefficient between *lineage distances* and *expression distances*, stratified by the *context distances* as follows:

1. We separate all the values of the *context distances* taken over time into 10 bins $\{C_1, C_2, ..., C_{10}\}$. The bins are redefined for each time point. For a given bin $C_i$ which is represented at that time point, we compute all the *lineage distances* and the *expression distances* and derive a value of Pearson correlation $r^{lineage}_{data}(C_i)$. We then obtain 10 different values of $r^{lineage}_{data}(C_i)$.

2. We perform the same calculation in the random null model at the same time point, which also gives different values of $r^{lineage}_{random}(C_i)$.

3. We compare the distributions of $r^{lineage}_{data}(C_i)$ and $r^{lineage}_{random}(C_i)$ using a Mann Whitney U test. The test gives us the p-value and the U statistics $U^{lineage}_{data}$ and $U^{lineage}_{random}$, which are related to the sum of ranks of the first sample, $r^{lineage}_{data}(C_i)$, and of the second sample, $r^{lineage}_{random}(C_i)$.

4. From $U^{lineage}_{data}$ and $U^{lineage}_{random}$ we calculate the rank-biserial correlation [25,26] to compare the two distributions $r^{lineage}_{data}(C_i)$ and $r^{lineage}_{random}(C_i)$. The rank-biserial correlation expresses the following: given all pairs of points formed by one point from $r^{lineage}_{data}(C_i)$ and one point from $r^{lineage}_{random}(C_i)$, in how many pairs $r^{lineage}_{data}(C_i)$ is larger than $r^{lineage}_{random}(C_i)$, or said in other words, what is the proportion of pairs of data points that are favorable to the hypothesis that the stratified correlations in *C. elegans* data are larger than those generated in the random null model. We use this rank-biserial correlation, which ranges from -1 to 1, as a proxy to measure the contribution of lineage-inherited factors, i.e. the mosaic mode. The rank-biserial correlation, with $n_{data}$ and $n_{random}$ being the number of values in the samples, here $n_{data} = n_{random} = 10$, reads: $rbc = \frac{2U^{lineage}_{data}}{n_{data}n_{random}} - 1 = 1 - \frac{2U^{lineage}_{random}}{n_{data}n_{random}}$. Because $U^{lineage}_{data} + U^{lineage}_{random} = n_{data}n_{random}$, $rbc$ can also be written, as our final expression for $W_{mosaic}$: $W_{mosaic} = \frac{U^{lineage}_{data} - U^{lineage}_{random}}{n_{data}n_{random}}$. $W_{mosaic}$ ranges from $-1$ to 1.

**Calculation of the contribution of the regulative mode $W_{regulative}$.** The contribution of the regulative mode $W_{regulative}$ is computed for each time point. We use the Pearson correlation coefficient between *context distances* and *expression distances*, stratified by the *lineage distances* as follows:

1. For a given *lineage distance* $l_i$ that is represented at that time point, we compute all the *context distances* and the *expression distances* and derive a value of Pearson correlation $r^{context}_{data}(l_i)$.

2. We perform the same calculation in the random null model at the same time point, which gives also different values of $r^{context}_{random}(l_i)$.

3. We compare the distributions of $r_{data}^{context}(l_i)$ and $r_{random}^{context}(l_i)$ using a Mann-Whitney U test. The test gives us the p-value and the U statistics $U_{data}^{context}$ and $U_{random}^{context}$ which are related to the sum of ranks of the first sample $r_{data}^{context}(l_i)$, and that of the second sample, $r_{random}^{context}(l_i)$.

4. From $U_{data}^{context}$ and $U_{random}^{context}$ we compute the rank-biserial correlations, as detailed in the previous section. These rank-biserial correlations lead to the definition of $W_{regulative}$ as $W_{regulative} = \frac{U_{data}^{context} - U_{random}^{context}}{n_{data} n_{random}}$ (where $n_{data}$ and $n_{random}$ are the number of values in the samples). $W_{regulative}$ measures to what extent the stratified correlations in the real *C. elegans* data are higher than in the random null model. $W_{regulative}$, ranging from $-1$ to $1$, is our proxy to measure the contribution of the space-dependent factors, i.e. the regulative mode.

**Calculation of the intrinsic range of variation in the random null model**

The $W_{mosaic}$ and $W_{regulative}$ are considered reliably high if they exceed the intrinsic variation range of the random null model. To define this intrinsic variation range, we perform the calculation of rank-biserial correlation that leads to $W_{mosaic}$ and $W_{regulative}$, but this time, the two distributions that are being compared are no longer the stratified correlations from the *C. elegans* data at time $t$ and the stratified correlations from the random null model at time $t$. Instead, the two distributions are the stratified correlations from the random null model at time $t_1$ and the stratified correlations from the random null model at time $t_2$, for all pairs $(t_1, t_2)$. This gives for the mosaic and regulative mode: $rbc_{mosaic}(t_1, t_2) = \frac{U_{random}^{lineage}(t_1) - U_{random}^{lineage}(t_2)}{n_{random}(t_1) n_{random}(t_2)}$ and $rbc_{regulative}(t_1, t_2) = \frac{U_{random}^{context}(t_1) - U_{random}^{context}(t_2)}{n_{random}(t_1) n_{random}(t_2)}$ We compute these values for all pairs of time points and we derive the mean and standard deviation of these values $< rbc_{mosaic} >_{(t_1, t_2)}$, $std(rbc_{mosaic})_{(t_1, t_2)}$ and $< rbc_{regulative} >_{(t_1, t_2)}$, $std(rbc_{regulative})_{(t_1, t_2)}$. The $W_{mosaic}$ and $W_{regulative}$ are considered to exceed the intrinsic variation range of the random null model if they lie above one standard deviation from the mean value.

**Octants definition**

The octant analysis provides a way to jointly examine the three types of distances linked to potential causal influence on cell state: *lineage distance* (mosaic influence), *context distance* (regulative influence), and *spatial network distance* (potential positional information such as in a morphogen gradient). For a given time point, each pair of cells is positioned as one data point in a 3D space, with coordinates given by the *lineage distance*, the *spatial network distance* and the *context distance* between the two cells. We separate this space into 8 bins (or octants) using the mid-point between the minimal value and the maximal value for each axis, as shown in Fig 3C. Thus, for each of the three distances, the mid-points classify cell pairs as "close" or "distant". This categorization yields eight possible configurations, or octants as illustrated in Fig 3. For example, the blue octant includes cell pairs that have low *lineage*, *spatial network* and *context distances*. This allows us to examine how cell pairs are distributed across octants and identify those with low expression distances.

We chose to use the mid-point between the minimal and maximal value as threshold, and not median or mean, because the shapes are very variable from one type of distance to another and from one time point to another, as shown in S5 Fig. Indeed, the mid-point threshold, although more prone to depend on outliers, is the method that is less prone to treat differently the different time points and different types of distances. Because the *lineage distances* and *spatial network distances* are integers, some data points might fall exactly on the midpoint values, and we arbitrarily assign these data points to the lower octant.

**Rearrangement rate**

For each time point and for each cell, the rearrangement rate aims to assess to what extent the neighbors at time t are the same as the ones at time t-1. The rearrangement rate is computed by comparing the list of neighbors at time t and at time t-1 with the Jaccard distance $j = 1 - \frac{|\mathcal{N}_t \cap \mathcal{N}_{t-1}|}{|\mathcal{N}_t \cup \mathcal{N}_{t-1}|}$ where $\mathcal{N}_t$ and $\mathcal{N}_{t-1}$ are the sets of neighbors of the cell of interest at time

t and time t-1. In case a cell among the neighbors has divided into two daughter cells between time t-1 and time t, the two daughter cells are considered as two different cells, and both are considered different from the mother cell.

## Results

**Quantification of mosaic and regulative contributions, based on stratified correlation compared to a random null model, reveals diverse modes of cell specification in major *C. elegans* tissues.** The detailed definitions of $W_{mosaic}$ and $W_{regulative}$ can be found in Materials and Methods.

In Fig 2, blue dots represent $W_{regulative}$ values; dotted blue lines indicate the range of intrinsic variation of the random null model (Materials and Methods). Filled dots mark significant *p*-values. Thus, a significant regulative contribution is inferred when filled blue dots exceeds the upper dotted line (see Materials and Methods). The same interpretation applies to the mosaic mode, shown in red.

To validate our approach, we applied the same analysis to an artificial lineage-defined expression model in which expression profiles depend solely on lineage position (Materials and Methods). As expected, in contrast to the real *C. elegans* data, this model shows very few significant blue dots (Fig 2F), indicating the absence of a regulative mode and confirming the significance of the significant values observed in the real data.

We computed $W_{mosaic}$ and $W_{regulative}$ separately for each tissue, assigning tissue identity to any cell lying on a genealogical line leading to a terminal cell of that tissue, labelled by Ma et al. [10]; most cells, and especially the earliest, thus belong to multiple tissues, the extreme case being the first cell P0, which belongs to all tissues.

Regulative mode contribution varies by tissue and time. In intestine, $W_{regulative}$ remains low. In the other four tissues, the regulative mode contribution becomes significant from a specific developmental time. This period starts at different time points depending on the tissue. In neurons and skin, this period starts at 86 min and 15 s and 88 min and 45 s while in muscle and pharynx, this occurs later: 170 min for muscle and 125 min for pharynx.

The mosaic mode contribution persists throughout most development, except at the end of skin development. Although $W_{mosaic}$ is initially not significant (likely due to the small number of cells), it generally exceeds the random null model range and later becomes significant in all tissues except skin, where it drops after 198 min 45 s.

Because $W_{mosaic}$ and $W_{regulative}$ do not sum to a constant, they cannot be interpreted as complementary weights; both may be high or low at the same time. Thus, they do not provide a completely disentangled description of both contributions. Also, they summarize all the cell pairs at a given time point into a single value. We complement these metrics with a joint analysis of all four distances: *lineage*, *context*, *spatial network*, and *expression*, to highlight specific populations of cell pairs of interest, and enrich the interpretation of the metrics.

**Clonally-related cell pairs exhibit similar expression profiles.** When distributing all cell pairs in the octants (as defined in Materials and Methods), we observe that three octants are nearly empty, as shown as uncolored octants in Fig 3C. on average, only between 0.3 and 0.7 % of pairs fall into these octants. This is expected, since cells that are closely related in lineage cannot be far in the spatial network, and cells that are close in the spatial network cannot be far in terms of *context distances*.

For each of the remaining five octants, shown as colored octants in Fig 3C, we can measure the percentage of cell pairs in this octant, as well as the mean *expression distance* for these cell pairs.

These five configurations represent (i) in blue: *clonally-related* cells which just divided from a common ancestor, still close in the spatial network and in terms of context, (ii) in orange: *context and spatial network related* cells that are close in context and in spatial network, while being distant in lineage, (iii) in red: *context-only related* cells that are close in context while being both distant in lineage and in spatial network (this configuration is a particularly interesting illustration of the regulative mode), (iv) in green: *spatially-close unrelated cells*, that are close in the spatial network without being close in lineage nor context, (v) in purple: *unrelated cells* that are close in none of the distance types.

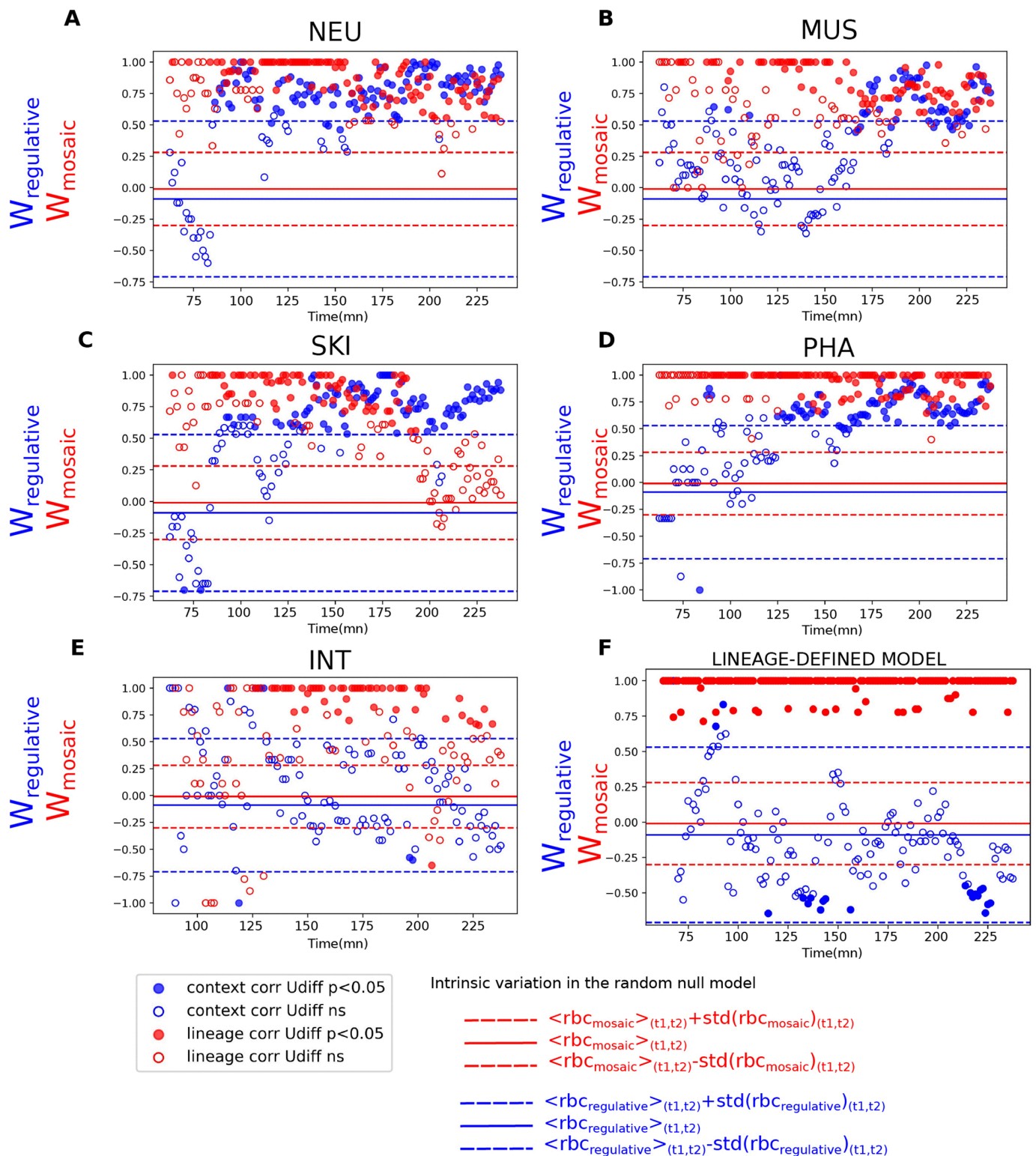

**Fig 2**. **Lineage (red) and context (blue) contribution to the various tissue types determination in _C. elegans_ and in the lineage-defined model.**
**A-E)** Lineage contribution is high in all tissues except for the end of skin determination (C). In neurons and skin determination (A and C) the mosaic contribution starts at 86 min 15 s and 88 min 45 s whereas in muscle and pharynx (B and D), it starts later at 170 min and 125 min, and it never is high in intestine determination. **F)** In the lineage-defined model, $W_{mosaic}$ is consistently high and $W_{regulation}$ remains low, as expected.

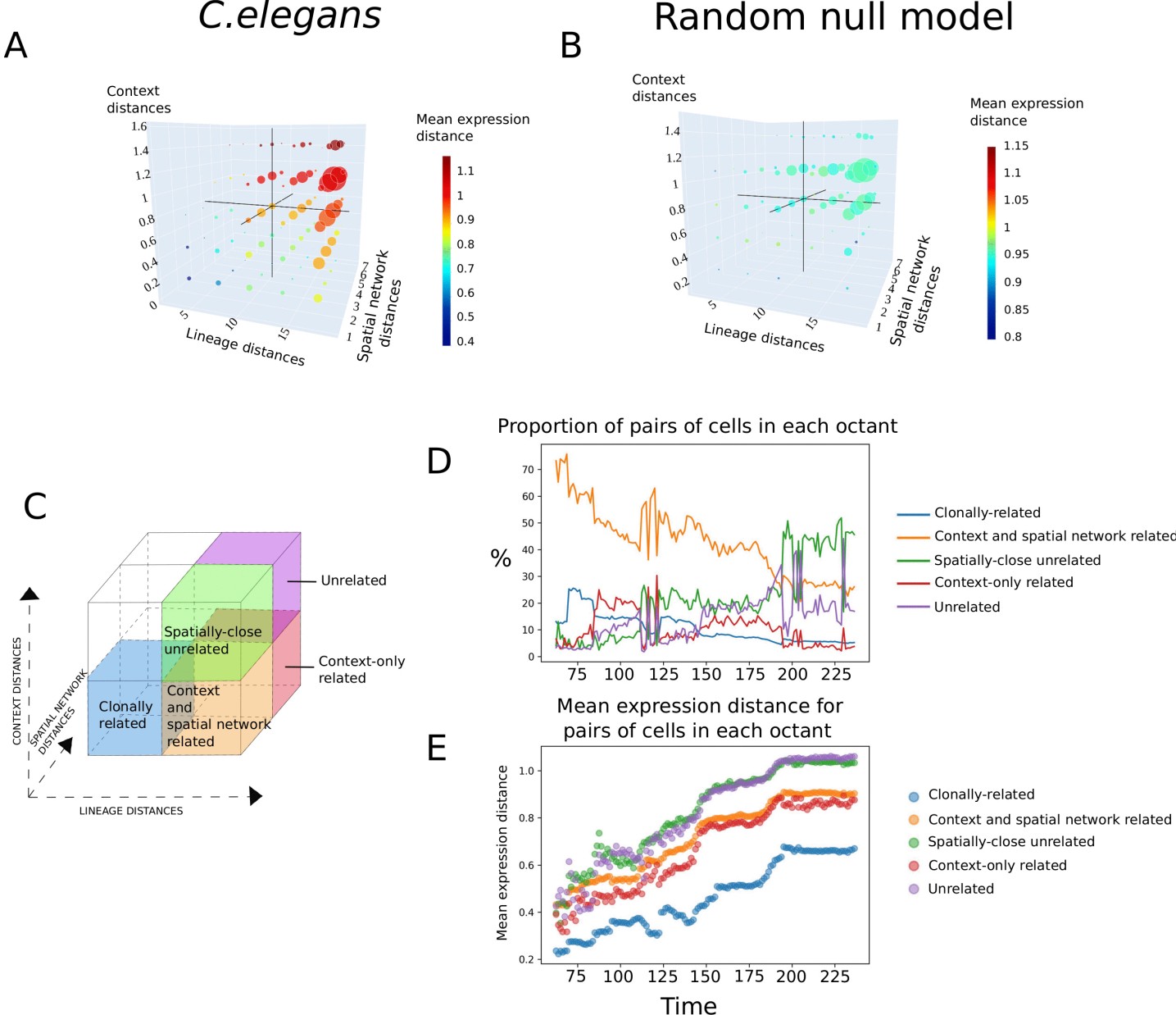

**Fig 3**. **Octant analysis reveals a population of cell pairs that are close only in context distance, and exhibit lower expression distances compared to unrelated cells in *C. elegans* A-B) Prior to octant analysis, 3D scatter plots show the distribution of all cell pairs at the final time point.** Axes represent *lineage distance*, *spatial network distance* and *context distance*. Cell pairs are aggregated using a binning of 5 along each axis. Each colored dot represents a bin, its size represents the number of cell pairs in the bin, and its color represents the mean *expression distance* within the bin. The black lines delineate the octants. This plot is shown for *C. elegans* data in A, and the random null model in B. **C)** Schematic representation of the octants definitions. Pairs "close" along an axis fall below the midpoint. **D)** Proportion of cells in each octant in time **E)** Mean *expression distances* in each octant in time

We had previously shown that $W_{mosaic}$ is generally high in almost all tissues and all developmental times as shown in Fig 2, suggesting that the mosaic mode plays a dominant role. The octant analysis refines this interpretation. The cell pairs which are close in lineage, referred to as clonally-related cells (in blue), have very low *expression distance*, in

comparison to all the other octants (Fig 3E). This confirms, alongside the high $W_{mosaic}$ values, that lineage-inherited factors play an important role. However, these clonally-related cell pairs are also close in *spatial network* and in *context distances*, meaning there is no distinct population of cells in which the influence of lineage can be fully separated from potential context effects.

**A population of cell pairs suggests disentangled effect of the context.** The effect of the regulative mode is expected to be strongest in context-only related octant (in red), which contains cell pairs that share similar context while being distant in both spatial network and in lineage. Fig 3D shows that this octant accounts for 10-30% of cell pairs between 62 min 30 s and 237 min and 30 s. As shown in Fig 3E, the *expression distances* within this octant (red) are lower than for unrelated cell pairs (green and purple). This indicates a subpopulation in which the effect of context on expression can be isolated from both the lineage and absolute spatial position.

We also observe that cell pairs in context-only related octant (red) and the context and spatial network related octant (orange) show similar mean expression distances. The similarity in their expression dynamics further suggests that spatial proximity alone, without shared context, is not a major driver of expression similarity.

By contrast, in both the lineage-defined model and the random null model (S4 Fig), the context-only related octant contains only a small fraction of cell pairs, and its mean *expression distance* is not lower than that of the unrelated cells (green and purple).

This octants analysis can also be done for each tissue separately, which gives additional insights as shown in S3 Fig.

**Proliferation and spatial rearrangement contribution to $W_{regulative}$ metric.** In the case of the skin specification, where Fig 2 shows a strong late-time $W_{regulative}$ signal exceeding $W_{mosaic}$, we examine how cell division and cell rearrangement contribute to this apparent regulative mode specification.

Fig 4A aligns $W_{mosaic}$ and $W_{regulative}$ with the rearrangement rate and the number of cells, in skin development over time, highlighting two intervals of interest:

1. Division-dominated phase (between dashed lines). A steep increase in cell number indicates a burst of divisions. In this interval, $W_{regulative}$ remains high, whereas $W_{mosaic}$ progressively loses significance. This suggests that newly born cells acquire expression profiles more strongly shaped by their local context than by lineage history, consistently with fate acquisitions through the regulative mode.
2. Rearrangement-dominated phase (after the right dashed line). Cell number reaches a plateau, indicating that no new divisions occur. Yet, $W_{regulative}$ continues to rise. This increase is then driven solely by cellular rearrangement, which rate (detailed in Materials and Methods) remains between 10% and 20%, across all periods without divisions. This suggests the presence of a sorting process placing similar cells in similar contexts. The timing coincides with late gastrulation and the onset of epidermal movements, beginning with closure of the ventral cleft which starts at 230 min [27].

For comparison, muscle development lacks a clear cell number plateau, reflecting more asynchronous cell divisions. The rearrangement rate stays in a constant range. In this case, our approach cannot distinguish between contributions of proliferation and rearrangements.

**An example of genealogically distant cells converging to the same fate following distinct trajectories.** In the previous section, as shown in Fig 3, we identified a population of cell pairs (those in the context-only related octant) that share similar context and expression profiles despite being distant in both lineage and spatial network. We now examine one such pair, observed at the final time point (237 min 30 s): ABplaapppp (green) and Cpappd (red) (in Fig 5). These cells are far apart in the lineage tree, and their genealogical lines diverge at the very first division, when the zygote P0 divides into AB and P1, with ABplaapppp deriving from AB and Cpappd from P1 (Fig 5B).

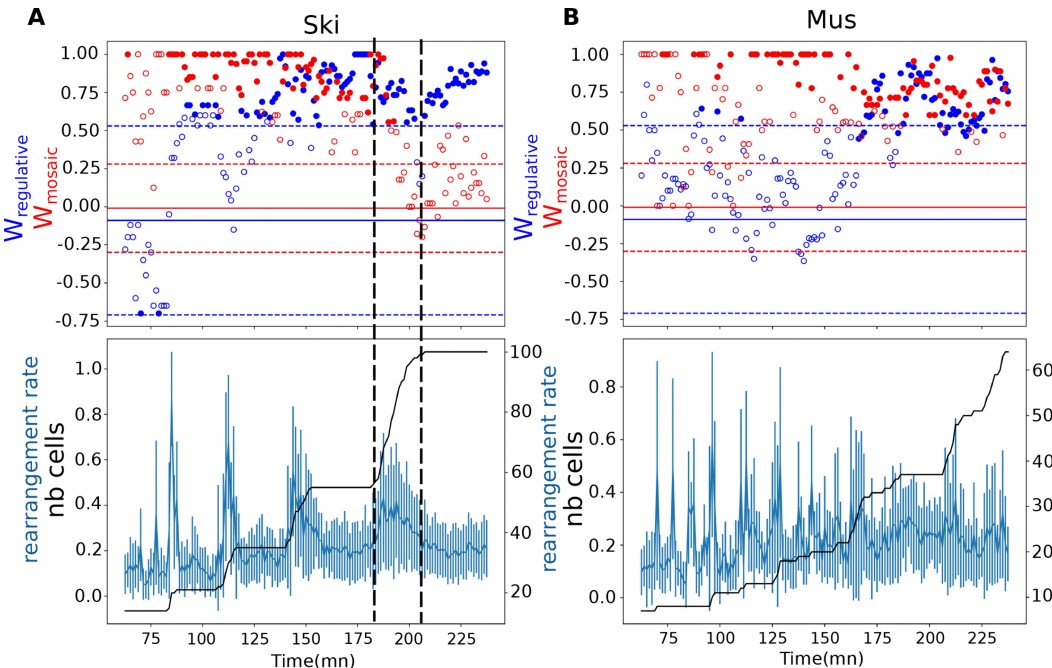

**Fig 4**. **Relationship between** $W_{mosaic}$ **and** $W_{regulative}$ **metrics, and cell proliferation and cellular spatial rearrangement. A)** Top panel shows the lineage and context contributions measured by $W_{mosaic}$ and $W_{regulative}$ in skin tissue. The bottom panel, with shared time point x-axis, shows the rearrangement rate in blue and the number of cells that belong to skin cells lines in black. The vertical dotted lines delimit the last tracked division round. The daughter cells generated during this round are context determined according to our measure (significant context $W_{regulative}$ in blue and non significant lineage $W_{mosaic}$ in red). After this round of divisions, whereas no new cells are generated, the context $W_{regulative}$ keeps increasing, suggesting sorting by cellular spatial rearrangement. **B)** For comparison, in muscle cells, the division rounds are more asynchronous and we cannot distinguish proliferation and rearrangement contributions.

Fig 5 traces the sequence of cells along each of the two genealogical lines. The spatial trajectory consists of the successive physical locations of the cells of the genealogical line (Fig 5C), while the expression trajectory refers to their positions in high-dimensional expression space (Fig 5A).

Their spatial trajectories diverge markedly, with the spatial network distance between them reaching up to five edges (Fig 5C and 5F)). The expression trajectories diverge too, before ultimately converging to a skin identity, in different regions of the embryo: Cpadppd (in red) on the posterior-right and ABplaapppp (in green) on the left-ventral part.

Interestingly, their *expression distance* (normalized in time to account for the overall divergence of expression as differentiation proceeds) drops at 193 min and 45 s. This convergence in expression is concomitant with the progressive drop in normalized *context distance* (Fig 5D and 5E) from 170 min to 193 min and 45 s. This observation is consistent with a regulative mode specification.

## Discussion

By jointly analyzing spatial positions, lineage information, and expression at single-cell resolution, we developed a methodology to quantify the respective contributions of mosaic and regulative developmental modes. Using *C. elegans* as a model, we found that both modes operate, at different times and in different tissues: the regulative mode acts earlier in neuron and skin specification than in pharynx and muscle, and is not detected in intestine development within the studied time window. The mosaic mode is present across all tissues and stages except at the end of skin specification. We identify a subpopulation of cell pairs in which the regulative mode appears to act in isolation, with no analogous cases for the

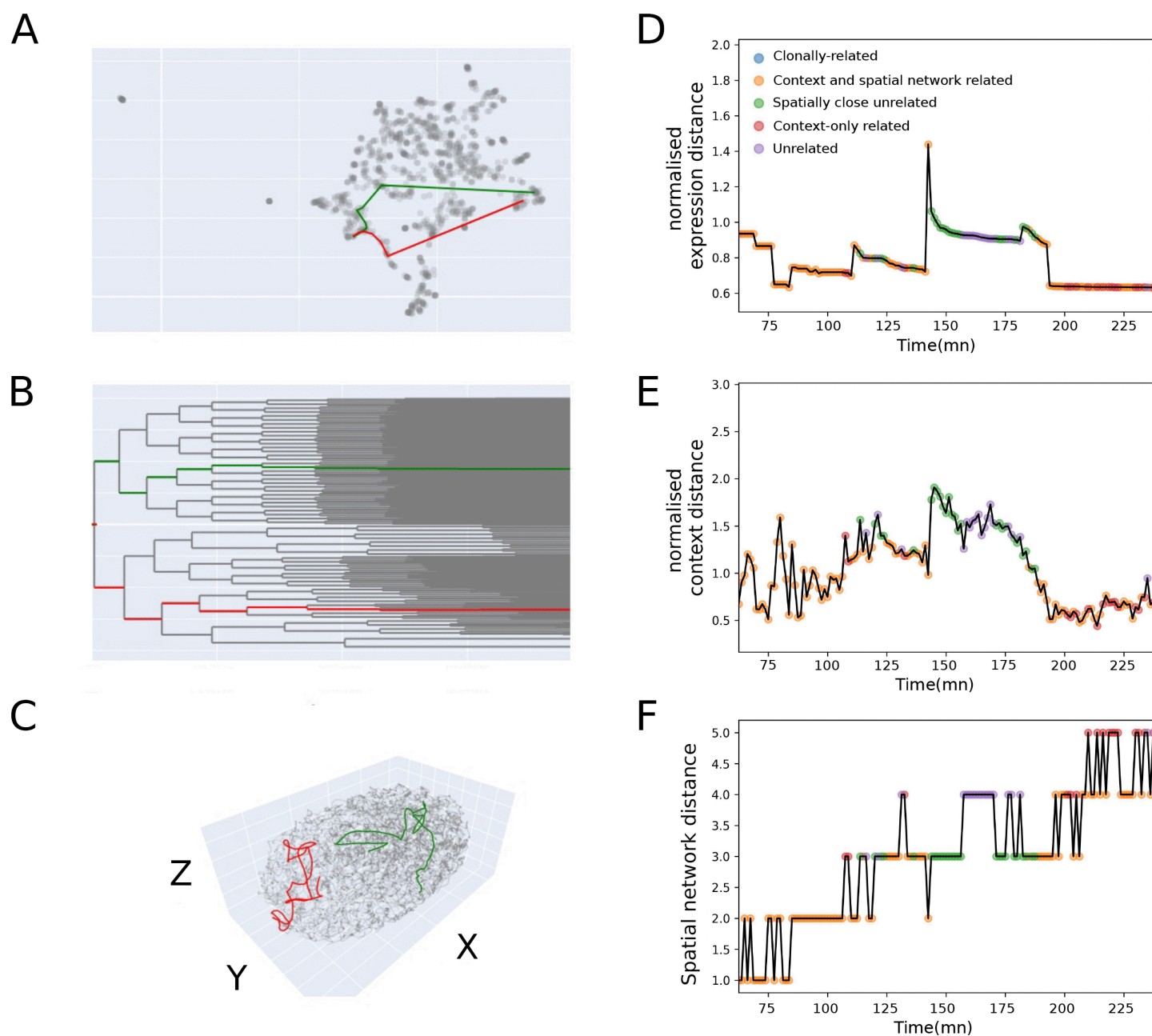

**Fig 5**. Illustration of two individual cell lines that lead to a pair of cells that are close in expression and in context, but far in lineage and in space, illustrating convergent fate acquisition **A)** UMAP representation of the expression of all cells generated across development. The red and green lines represent the trajectories that lead to the pair of cells. **B)** the trajectories in the lineage tree. **C)** the trajectories in 3D physical space **D)** *expression distance* between the two trajectories normalized by the mean *expression distance* between all the pairs of cells of each time point. This confirms the convergence of cells expression visible in the UMAP in A). The colors of the dots correspond to the octants defined in Fig 3 **E)** the *context distances* decrease at about the same time as the *expression distances* **F)** the *spatial network distances* increase. The pair of cells occupy different regions of the skin.

mosaic mode. Our results also suggest that correlations between context and expression may arise from cellular sorting, as cells with similar expression profiles may rearrange to occupy similar neighbourhoods. Finally, we highlight an example

of two distantly related cells converging to the same fate through distinct trajectories in physical and expression space, accompanied by a decrease in context distance.

Several refinements could improve this method. In our current approach, the context of a cell is defined as the mean expression of its neighbors, inferred from a simple Delaunay network connecting cell centers. First, this could be replaced by using experimentally determined neighbors, available through a morphological atlas of *C. elegans* [24], potentially weighted by contact areas, which can determine the strength of cell-cell interactions [22]. Then, more sophisticated strategies could be explored to integrate neighborhood information than a weighted mean, such as graph neural network based models [28], or latent factor approaches [29]. Additionally, other measures of cell state could be incorporated. For instance, including transcriptional states from the comprehensive single-cell transcriptomic atlas [9] would provide complementary information. However, this atlas currently includes a substantial number of transcriptomic profiles that cannot be uniquely assigned to lineage positions at single-cell resolution due to left–right symmetry in the embryo [30], limiting the possibility of unambiguous cell-to-profile mapping in *C. elegans*.

Another limitation of our method comes from using correlations between expression distances and context distances among cells existing at the same time point as a proxy for regulative mode. This method cannot distinguish whether the high correlation between these distances is due to similar neighborhoods inducing similar fates, or whether similar cells have sorted to similar neighborhoods as suggested in previous works [31,32], and as we observe at the end of skin development. A stronger causal argument for regulative mode would require showing that small *context distances* precede small *expression distances.* Approaches such as Granger causality [33] could in principle infer causality from time-series data. However, in our dataset expression is measured only once per cell cycle, limiting the temporal resolution needed for such analyses.

The specificity of our method is that it operates on pairwise distances between cells, rather than individual parameters. This reduces problem complexity by lowering dimensionality and focusing on the interactions between cells. The strategy is conceptually related to dimensionality reduction and manifold learning methods such as MultiDimensional Scaling or Isomap [34], and offers improved robustness to noise. Below, we review related work and highlight how our approach differs.

Quantifying the relative contributions of mosaic and regulative modes is closely related to previous work on assessing the roles of signal and noise in development [35–37]. Cell diversification is often conceptualized using Waddington's landscape, where cells "roll" down hills into valleys representing distinct fates. Divergence from the same initial state can arise through several mechanisms: (i) stochastic noise shifting slightly initial positions (ii) cell-cell signaling deforming the landscape, or (iii) asymmetric inheritance of factors during cell division [16]. Comparing (i) and (ii) corresponds to a signal-versus-noise analysis, which usually requires knowledge of the underlying gene regulatory networks. In *C. elegans*, development is highly stereotypical with invariant lineage, so noise is minimal. We therefore focus on comparing cell–cell signaling (ii) with lineage-inherited factors (iii), without inferring regulatory networks, using statistical inference from observed data alone.

Other studies have used statistical inference to investigate transmission of information across generations. These studies jointly analyze lineage trees and some phenotypic features. For example, Tran et al. [38] detect recurrent lineage motifs to identify progenitor states, and assess how these motifs constrain final cell-type proportions. Natesan et al. [39] quantified variability in the *C. elegans* lineage across individuals, considering not only tree structure but also cell cycle lengths using tree-edit distance [40]. Hicks et al. [41] have examined how features such as gene expression, cell size, or division time vary along the lineage, identifying where they stabilize or remain flexible across generations. Our work is also built on a statistical approach to investigate transmission of information across generations, but additionally incorporates spatial relationships as a second source of influence on cell phenotype.

Finally, we connect our work to the previously described algorithmic complexity of lineage trees [42]. In *C. elegans* in particular, it was shown that the complexity is lower than expected by chance, yet not optimal. A proposed explanation for

this sub-optimality is that development must not only generate the correct cell types, but also position these cells appropriately in the embryo. Our study reinforces the idea that developmental complexity arises from both lineage-based information flow, as previously measured [42], and neighbor-mediated cues, implying that an ideal measure of complexity (potentially optimized by evolution), should incorporate both aspects. In this regard, it would also be interesting to inquire about the evolutionary conservation of the mosaic and regulative modes of development across multiple species, starting for example with close species such as *C. elegans* and *C. briggsae* [43].

## Supporting information

**S1 Fig. Pearson correlation r between all pairs of measured distances.** The values are mildly positive. The raw correlations are hardly interpretable because of the mixed effects of lineage, spatial positions and context. This justifies to build a pipeline to disentangle the various sources of correlation.
(PDF)

**S2 Fig. Proportion of shared neighbors between two cells.** Two cells which are close in terms of spatial positions might share a certain proportion of their respective neighbors. When this is the case, the context distance between them will be low just because most of their neighbors are common. Here we plot A) the proportion of shared neighbors between two cells with respect to the physical distance between them. The physical distance is defined as the length of the shortest path between them in the Delaunay graph of all cells nuclei positions. The proportion of shared neighbors is calculated as the Jaccard index between the respective sets of neighbors of the two cells. B)The average and standard deviation of Jaccard index for all pairs at a given time point.
(PDF)

**S3 Fig. Octant analysis of each tissue separately in *C. elegans* data.** Each row shows the same analysis as for Figure 3, performed by retaining only the pairs of cells that belong to a given tissue, in the sense that they belong to lines that lead to terminal cells labeled as belonging to this tissue.
(PDF)

**S4 Fig. Octant analysis for the artificial models.** Same analysis as in Fig 3 performed on the lineage-defined model and the random null model.
(PDF)

**S5 Fig. Distributions of each of the four types of distances.** Distributions of each of the four types of distances at different time: lineage distances, context distances, expression distances and physical distances. The distributions are asymmetric and their shapes vary with time.
(PDF)

**S6 Fig. Tissues labels visualization.** The spatial distribution of cells belonging to each labelled tissue at t = 237 mn. At this time step, which is the last one in which the spatial positions are tracked, the cells are not yet all fully differentiated so that some cells may appear more than one tissue if their daughters have different tissue types. Cell types visualization with UMAP.
(PDF)

## Acknowledgments

We thank Nicolas Levernier, Vincent Bertrand, Pierre Recouvreux, Khulganaa Buyannemekh, Raphaël Clément, Charlotte Rulquin, Florence Bansept, Dominic Skinner for fruitful discussions. We used OpenAI. (2021). ChatGPT: Large-scale generative models for natural language conversations. https://platform.openai.com/models/chatgpt and Deepl Write for text editing.

## Author contributions

**Conceptualization:** Solène Song, Paul Villoutreix.

**Funding acquisition:** Paul Villoutreix.

**Investigation:** Solène Song.

**Methodology:** Solène Song, Paul Villoutreix.

**Writing – original draft:** Solène Song, Paul Villoutreix.

**Writing – review & editing:** Solène Song, Paul Villoutreix.

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
