## [Decision Letter · Decision Letter 0]

24 Sep 2024

Dear Dr Villoutreix,

Thank you very much for submitting your manuscript "Assessing the Relative Contributions of Mosaic and Regulatory Developmental Modes from Single-Cell Trajectories" for consideration at PLOS Computational Biology.

As with all papers reviewed by the journal, your manuscript was reviewed by members of the editorial board and by several independent reviewers. In light of the reviews (below this email), we would like to invite the resubmission of a significantly-revised version that takes into account the reviewers' comments.

We cannot make any decision about publication until we have seen the revised manuscript and your response to the reviewers' comments. Your revised manuscript is also likely to be sent to reviewers for further evaluation.

Sincerely,

Adriana San Miguel

Academic Editor

PLOS Computational Biology

Marc Birtwistle

Section Editor

PLOS Computational Biology

Reviewer's Responses to Questions

**Comments to the Authors:**

Reviewer #1: In this article, the authors propose a new computational method to evaluate the contribution of mosaic and regulative modes to the development of an organism and apply it to an extensive dataset of C. elegans development. The authors first propose a measure for the contribution of the mosaic mode at a particular developmental time based on the correlation between cell-cell lineage distance and cell-cell expression distance, while the regulative mode contribution is measured as the correlation between cell-cell context distance and cell-cell expression distance. The authors leverage two extensive datasets of C. elegans development jointly containing, for each cell, information about their lineage, spatial positions and gene expression in time and use these measures to evaluate the contribution of mosaic and regulative modes during the development of different C. elegans tissues. The authors find that while the development of some tissues, such as the intestine, develop mainly through a mosaic mode, other tissues, such as the skin, have a higher mosaic contribution, particularly at the end of their development. The authors also study the effects of context, lineage, and spatial position on gene expression distance as well as the correlation between the changes in the modes and the proliferation and spatial rearrangements.

I believe there is a convincing motivation for the authors’ study, and the two rich datasets used in the study form a great context to test and leverage their proposed method. I think their computational method is mostly sound, and their claims seem supported by their analyses. A possible caveat of their method, which the authors mention in the Discussion section, is that the authors evaluate the contribution of each mode by a measure of correlation between the gene expression distance at time t, and the lineage or context distance at time t. However, one could imagine that the effect of the context, lineage, or spatial position in the gene expression, and particularly cell fate, could take longer than 75 seconds (I am not sure if this is true in C. elegans in particular, but it is in other organisms). I still think, however, that their approach is relevant and a first step toward a more general method.

Having said this, I have several concerns about the current state of the manuscript, which I detail below:

Major comments

1.- The authors utilize two datasets that record spatial position and gene expression data every 75 seconds for 6 hours, which I believe would result in around 288 time points. However, in the Datasets section of the paper, lines 110 and 112, figures, and the rest of the manuscript, the authors mention that the time points range from t=0 to t=190. Could the authors clarify how they got to that number? I believe this is important, as all the conclusions depend on this dataset.

2.- I believe the text and organization of the manuscript need some revision:

- I would suggest the authors revise the writing and figure labels as I have seen quite a few typos (two “the” on line 17, “it’s” instead of “its” on line 106, “build” instead of “built” on line 149, “replacing inverting” on line 177,…).

- In the second subsection of the Results section, the introduction of the 8 quadrants seemed a bit abrupt, while the third subsection introduces this quantification in great detail. Also, in the second subsection, the authors mention “unrelated pairs of cells” (line 348), which is not defined until the third subsection. Have the authors considered changing the order of these subsections?

- In lines 381-382 the authors write “This analysis by configuration can also be done for each tissue separately, which gives additional insights as shown in supplementary.” I haven’t been able to find any supplementary information attached to the submission, could the authors clarify what this supplementary is?

- In lines 415-417 the authors briefly mention the behavior of muscle cells as opposed to such of skin cells in how proliferation and spatial rearrangements change the modes of specification. Could the authors expand on how the differences in divisions and rearrangements between muscle and skin cells translate into different modes and whether their model can give an explanation for this difference? Do the authors suggest that exclusive regulative mode needs for high proliferation and rearrangement rate?

- In Figure 2, the continuous and dashed lines are said to be the intrinsic variation in the shuffled toy model, do the authors mean the null or the lineage toy model?

- The title of Figure 3D appears cut.

- Legend of Figure 3C, could the authors clarify what “the tracks in 3D space” mean? Are they the spatial coordinates of the two cells in time?

3.- When the authors study the effect of the context, independently of lineage or spatial relationships, they state that Figure 3 shows that the red quadrant, where the cells have similar context but are distant in space and lineage, “comprises between 10 and 50% of cells along time points between t=50 and t=190” (lines 337-339). However, in Figure 3C, if I understand it correctly, the red line is always below 30%. Could the authors clarify this inconsistency?

Minor comments

1.- The authors measure rearrangements as the Jaccard distance between the list of neighbors at time t and at time t-1. How do the authors deal with cell divisions where the neighbors at time t and t-1 are different cells? I might not have understood how they measure rearrangements, so I would really appreciate if they could expand on their method.

2.- When the authors compare the contributions of each mode in each tissue, they explain that they “consider as being part of a tissue all the cells that belong to tracks leading to the leaves that are labeled as this tissue”, lines 301-303. However, all the cells in the organism come from the first single cell, and also, there might be progenitors that give rise to two tissues eventually. How do the authors deal with this issue? Do the authors start the quantifications shown in Figure 2 from the first cell quantified or from the first progenitor that exclusively gives rise to that tissue?

3.- When the authors study the effect of the context, independently of lineage or spatial relationships, they classify the data across quadrants using the mid-distance between the minimal and maximal value of each type of distance (lines 331-332). I was wondering if the authors have considered any other distance more stable to the presence of outliers, such as the median of each type of distance.

Reviewer #2: In this work, the authors studied the two modes of specifying cell types in development, i.e., mosaic and regulative modes. Specifically, they aim to quantify the respective contributions of the two models from single-cell data for C. elegans. By calculating some measured defined by the authors and correlation analysis, they demonstrate the co-existence of mosaic and regulative modes in development of C. elegans. Some specific conclusions include that in the skin during early development, the mosaic mode dominates while at later stages, regulative mode dominates.

Overall, although I appreciate the efforts to using single cell data to explore the cell fate determination problem, I didn't see much new insight from this paper from the perspective of cell fate decision mechanism. A major problem is that the authors show many different observations from statistics of data, but the underlying mechanism, especially the one related with cell fate decision, a core problem in systems biology, is mostly not explored.

1, The two modes proposed, mosaic mode and regulative mode, seem related to the two mechanisms in cell fate decision, i.e., noise driven or signal driven mechanism (Coomer et al. (2022) Cell Systems 13:83-102). Can the authors elaborate the relationship between their work and above two mechanisms?

2, The authors conclude that both modes (mosaic mode and regulative mode) could contribute to the development at various times in the various tissues. In the field of systems biology, it seems widely acknowledged that both noise and signal contribute to cell fate transitions (Li and Wang 2013 PLOS Comput Biol, Xue et al. eLife 12:RP88742 (2023)).

3, Usually, cell fate means cell type. i.e., similar cell fate (similar gene expression pattern in a high-dimensional gene expression space) should correspond to similar cell type. In current work, the authors show that two cells with similar cell fate could have very different lineage (Figure 5). Can the authors elaborate what this mean? Why is this observation reasonable? And what is the underlying mechanism for this?

4, Figure 5 shows that cell fate transition can go through different route for reaching similar endpoint. Similar conclusion has been predicted by attractor landscape model based on gene regulatory networks (Kang et al. Advanced Science, 8: 2003133 (2021).).

5, There are grammar issues in the manuscript, e.g., line 271 “...we will explicit only one of them...”, line 404, “...with their context than with with their lineage inheritance...”. Please check the expressions throughout the manuscript.

Reviewer #3: This paper tackles the problem of understanding expression patterns on cell lineage trees. Here the authors develop metrics to explain expression differences between pairs of cells in terms of certain covariates, in particular lineage (or kinship) distance and context distance. They define context distance to be the correlation distance between the mean expression profile of the direct neighbors of each member of the cell pair.

For each cell pair, there is thus a lineage distance, an expression distance, a context distance, and a physical distance (measured in terms of the Delaunay graph). The study attempts to remove errors of confounding by stratifying the data and conditioning on certain distance types.

I think the paper presents an innovative and original methodology. I believe it to be an important addition to the toolset addressing how we analyze expression patterns in cell lineage trees. The biological insights - for example that lineage ("mosaic") influences are largely dominant, with environmental ("regulative") influences sometimes appearing late - are helpful and in line with expectations for C. elegans.

The problem of quantifying expression patterns on cell lineage trees has been receiving increased attention. Several workers have studied distances/differences between pairs of cells. Although the current manuscript goes further in defining distances in lineage, context, and space, as well as expression, I do think it would be helpful if the authors place this work in the context of previous work in this area, such as (though not exclusive to) the following: Lineage motifs as developmental modules for control of cell type proportions, Martin Tran, Ahmad Askari, Michael B. Elowitz, Developmental Cell 59, 812-826 (2024); Novel metrics reveal new structure and unappreciated heterogeneity in Caenorhabditis elegans development, Gunalan Natesan, Timothy Hamilton, Eric J. Deeds, Pavak K. Shah, PLOS Computational Biology, 19(12): e1011733 (2023); Maps of variability in cell lineage trees, Hicks DG, Speed TP, Yassin M, Russell SM, PLOS Computational Biology. 15(2):e1006745 (2019)

Below are some detailed points:

p.3, line 92: The effect sizes W_mosaic and W_regulative are introduced here well before they are defined. I suggest giving a short description of them and cross-referencing the Methods section where they are defined.

p.4, line 110: Please specify the unit of time used. If t=190 is about 6 hrs (line 112), this suggests 1 unit of time is about 2 mins? Given its importance in the discussion, I would suggest the time be specified in hrs or minutes.

p.4, line 138: Why is the spatial distance specified by the (discrete) steps in a Delaunay graph? I would have expected the Euclidean distance, which is continuous, to be a more accurate measure of proximity. I am not suggesting this be changed, it would just be helpful to have a comment about why this distance measure was chosen.

p.5, line 161: The random walk model needs more explanation. x and v need to be defined.

p.5, line 177: The short cut phrase "...replacing inverting lineage and context" is unclear. Writing this statement out explicitly would be make the meaning more clear.

p.5, line 186: The context distances are stratified into 10 bins. Please be specific that (I assume) these 10 bins are created at each time point. I assume the bin sizes and bin positions vary over time as overall expression changes?

p.6, line 198: Why do the definitions of W_mosaic and W_regulative involve subtracting the U statistic of the randomly-permuted model? This does not appear to be a standard rank-biserial correlation so please provide justification or a reference.

p.6, line 224: When dividing the 3-dimensional space of lineage distance, physical distance, context distance into 8 regions, each region is termed a quadrant. This was confusing to me. I would suggest calling them octants since quadrants implies a division of a 2-dimensional space into 4 regions.

p.8, line 271: The phrase "...we will explicitly only one of them..." needs to be clarified.

p.8, line 274: In the sentence "We want to measure this strength independently of the lineage distance which is likely to be a confounding factor in this correlation" I would suggest replacing "independently of" with "after controlling for".

p.9, line 291: Reference is made to "W_regulative of the null model (by computing the U test between all pairs of time points within the null model)". How does this relate to the original definition that W_regulative ~ U_data-U_null?

p.10, line 338: "...between 10 and 50%". Doesn't the red line in Fig 3c only go up to ~30%?

p.10, line 367: Is "the percentage of cells in each configuration over time" actually a percentage of cell pairs? This refers to Fig 3c and its caption as well. I thought that the octants in Fig 3a were populated by points corresponding to cell pairs, not individual cells. I presume an individual cell could appear in multiple octants since it would usually be represented in multiple pairs.

p.11, line 403: "The new cells that just divided have adopted expressions that correlate more with their context than with their lineage inheritance." This sentence and the ones immediately after it were a bit confusing to me as it wasn't obvious that the context was causing the expression rather than the other way around. The later discussion about causality and time resolution was helpful but this paragraph could benefit from more explanation.

p.19, Fig. 3d: Does the fact that the red (context only) and orange (context and physical) lines almost overlap provide further evidence that physical proximity alone (without context) is not a major driver of expression?

**Have the authors made all data and (if applicable) computational code underlying the findings in their manuscript fully available?**

Reviewer #1: Yes

Reviewer #2: Yes

Reviewer #3: Yes

PLOS authors have the option to publish the peer review history of their article (what does this mean?). If published, this will include your full peer review and any attached files.

Reviewer #1: No

Reviewer #2: No

Reviewer #3: No
---

## [Decision Letter · Decision Letter 1]

23 Apr 2025

PCOMPBIOL-D-24-01234R1

Assessing the Relative Contributions of Mosaic and Regulatory Developmental Modes from Single-Cell Trajectories

PLOS Computational Biology

Dear Dr. Villoutreix,

Thank you for submitting your manuscript to PLOS Computational Biology. After careful consideration, we feel that it has merit but does not fully meet PLOS Computational Biology's publication criteria as it currently stands. Therefore, we invite you to submit a revised version of the manuscript that addresses the points raised during the review process.

Please submit your revised manuscript within 60 days Jun 23 2025 11:59PM. If you will need more time than this to complete your revisions, please reply to this message or contact the journal office at ploscompbiol@plos.org. Please include the following items when submitting your revised manuscript:

We look forward to receiving your revised manuscript.

Kind regards,

Adriana San Miguel

Academic Editor

PLOS Computational Biology

Marc Birtwistle

Section Editor

PLOS Computational Biology

**Additional Editor Comments :**

Please address the remaining concerns by reviewers two and three, in particular to points raised about clarification of the approach, as well as the sensitivity of test for significance to the specifics of the null model if the authors agree with this comment.

**Reviewers' comments:**

Reviewer's Responses to Questions

Reviewer #1: I appreciate the author’s efforts in addressing our concerns. The revised version of the manuscript has indeed improved. Nevertheless, I still have some reservations about the article.

First, the language and grammar of the article should be thoroughly revised. Some typos and grammatically incorrect sentences include but are not limited to:

• Line 33: “One the remaining question is…”

• Line 53: “Some studies other have focused on…”

• Line 120: “…can be responsible of…” should be “can be responsible for”

• Line 133: “…with reference to previously introduced random model” should be “with reference to the previously introduced random model”

• Lines 304–307 could be written more formally. For example, instead of “vary a lot,” one could write “are very variable.”

• Line 330: “the Pearson r” should be “the Pearson correlation r”

• Line 340: “Material and Methods” should be “Materials and Methods”

• Lines 449–450: “Figure 3 show” should be “Figure 3 shows”

• Line 458: “…is similar if the cells are close in physical space or not” could be changed to “independently of their physical distance”

• Line 483: “there is steep drop of the lineage” should be “there is a steep drop in the lineage”

I appreciate the effort in including the other reviewers’ suggestions to mention previous related work in the Introduction. However, I believe the Introduction needs further revision. For example, the authors argue in lines 41–42 that signal vs. noise quantification is not relevant to this study because “Here, in the case of C. elegans, which displays very stereotypical development with little variation, with invariant lineage, we expect little noise.” At that point in the text, though, the authors have not yet introduced their main goal or clarified that they are focusing on C. elegans.

I still do not understand the dataset completely. In lines 148–149, the authors note, “At the last time point t=190 (about 4 hrs), there are 380 cells. By this stage, the embryos have completed all but the last round of cell divisions, and have produced 762 cells, for a total number of 1341 different cells observed across the entire development.” Do the authors mean that embryo 13 has 380 cells tracked out of the 762 cells that an embryo would have at t=190, and that from t=0 to t=190, the dataset comprises a total of 1341 cell states?

In lines 182–183 of the Materials and Methods section, the authors define the context distance between two cells A and B as the correlation distance between the mean expression profile of A’s direct neighbors and the mean expression profile of B’s direct neighbors. However, the neighbors of a cell are not defined until lines 314–315. It would be clearer if the authors reordered these explanations.

In Figure 3C–E, the authors classify and describe various relationships between cells in the dataset. In particular, they focus on a subset called “context-only related” cells, which—according to line 437—is a key illustration of the regulative mode. They show in Figure 3D how the percentage of these context-only related cells changes over time, peaking between 62 and 190 minutes. However, Figure 2 indicates that the regulative mode is predominant at later times in all cell types, especially after 200 minutes. I am curious about how W_regulative relates to the percentage of context-only related cells. If my understanding is correct, a higher regulative contribution should correspond to a higher percentage of context-related cells. This would be consistent with observations in the Intestine—where cells are mostly clonally related, and the mosaic mode predominates. I would appreciate if the authors could explain these relationships further.

Regarding Figure S3, the percentage of clonally related cells in the Intestine is very high but sometimes drops to 50%. I am wondering what types of cells make up the remaining 50%.

Reviewer #2: .

Reviewer #3: I accept the responses and manuscript changes and recommend publication.

I do have a comment regarding the "Lineage-defined model" on page 6/23 (thank you for explaining it in more detail). I think there is wide scope for defining null models that exhibit purely mosaic variation. The Langevin-type equation the authors have chosen is a specific type of random walk on the lineage tree that has limited memory. One could imagine null models with longer-term memory that explain more of the correlations at large lineage distances. Do the authors think this could affect their conclusions? If so, I recommend the authors note this in the text by saying that the test for significance (for mosaic variation) is sensitive to the specifics of the lineage-based null model.

**Have the authors made all data and (if applicable) computational code underlying the findings in their manuscript fully available?**

Reviewer #1: Yes

Reviewer #2: Yes

Reviewer #3: Yes

PLOS authors have the option to publish the peer review history of their article (what does this mean?). If published, this will include your full peer review and any attached files.

Reviewer #1: No

Reviewer #2: No

Reviewer #3: No

**Figure resubmission:**
---

## [Decision Letter · Decision Letter 2]

15 Aug 2025

PCOMPBIOL-D-24-01234R2

Assessing the Relative Contributions of Mosaic and Regulatory Developmental Modes from Single-Cell Trajectories

PLOS Computational Biology

Dear Dr. Villoutreix,

Thank you for submitting your manuscript to PLOS Computational Biology. After careful consideration, we feel that it has merit but does not fully meet PLOS Computational Biology's publication criteria as it currently stands. Therefore, we invite you to submit a revised version of the manuscript that addresses the points raised during the review process.

Please submit your revised manuscript within 60 days Oct 15 2025 11:59PM. If you will need more time than this to complete your revisions, please reply to this message or contact the journal office at ploscompbiol@plos.org. Please include the following items when submitting your revised manuscript:

We look forward to receiving your revised manuscript.

Kind regards,

Adriana San Miguel

Academic Editor

PLOS Computational Biology

Marc Birtwistle

Section Editor

PLOS Computational Biology

**Additional Editor Comments:**

Dear authors: as you can read from reviewer 4, there are still some concerns with the methods section, in particular the clarity, rigor, and precision. It could be helpful if the text is read by an expert with mathematics and embryo development background to ensure the revised version is adequate for publication.

**Reviewers' comments:**

Reviewer's Responses to Questions

**Comments to the Authors:**

Reviewer #1: I appreciate the authors' efforts in revising the manuscript and responding to my questions. I accept the responses and the revisions and recommend the manuscript for publication.

Reviewer #4: I do appreciate the efforts of the authors to answer my previous comments. However, I still think that the method description needs clarification: for the moment, part of the method text is still quite difficult to read (the formalisation lacks rigor and precision), and I would not be able to reproduce the authors method independently as the description still contains ambiguities and notational issues that blurs the overall approach (which otherwise remains original and interesting).

Here are comments to help identifying and addressing these issues (they are mainly related to the method section).

p2, L6-7: subtitles in the introduction are not usual. At least the first title is very strange and poorly introduced by the two first sentences. it sorts of restates the title of the paper slightly differently.

p 3, L70-74: About the core assumption: the closer the cells in the lineage tree, the higher their expression profile.

In a Mosaic-based embryo lineage tree that develops two symmetric sides (Left-Right), two cells with identical fates taken respectively in different sides Left and Right of the embryo, will be far in the lineage tree (their closest common ancestor is close to the zygote cell). However, their expression neighborhood will be similar.

This is thus a situation where in the Mosaic development, two cells have a high lineage distance and same expression. However, according to the core assumption of the paper, they should have distant expression profile, which is obviously not the case.

Should the assumption of the authors remove first symmetries in the embryo ? (i.e. their assumption holds if L/R symmetry is discarded) ? Are these situations rare, when correlations are considered ? I think the author should develop the intuition behind their core assumption and discuss these questions to justify their choice.

p4 L115-188: the notion of "weight of factors" is not defined and is used intuitively. At this point, without having read the rest of the paper, it is very unclear. The weight in which measurement process ? which factors ? are we still referring to the distances that are introduced before ?

p4: First paragraph. Why are Wmosaic and Wregulative values between -1 and 1 ? This should be explained as one naturally expect these W to be between 0 and 1: 0 meaning no contribution at all, 1 meaning full contribution. I understand latter in the text, that these factors are linked with some notion of correlation (which is then properly defined even farther in the text). I strongly advise the authors to consider the order in which they introduce concepts and definitions. This order should be so that one does not need to read the rest of the text to understand a definition of a statement. Please check this issue, not only in this particular instance, but throughout the method text. The precision of all these definitions is key and should be made at the usual scientific standards: for the moment, and even after the revision, although I am familiar with these types of formalizations, I am really struggling to understand clearly what was precisely done from the description in the text. See also below.

p5: L141-158: The datasets section does not mention which embryos where used. Most of the related information is badly placed in the next section (spatial position). Is it 1 or several embryos that were used in this study ? According to L152, I understand that this is one (embryo 13). This should be in the dataset section.

p5: L145: "partial expression profile": please, explain in which way this is partial. Is partiality an issue at all for the conclusions ? why ? This should be discussed

p5: L177: "network" should probably be replaced by "tree-graph", more standard and adequate in this context.

p6: L181: According to its name, the function ShortestPath() returns a path, not a length. I therefore suggest to rewrite "d(A,B) = ShortestPath(A,B)" as d(A,B) = length(ShortestPath(A,B) ). Same at line 193.

p6: L184: you should mention explicitly that x_A and x_B are vectors of reals with a certain dimension, and which one?.

The notation for mean value should be defined here (this is another instance of the ordering problem mentioned above). This should mention the set over which the mean value is taken: for the moment, I don't understand precisely what is <x_a> : is it an average over all the "same cells" in a set of embryos ? Or is it an average over a specific tissue type in the same embryo ? I bet this is the second, but please realize that this is not defined at all in your definitions and that this is critically lacking at this moment of the text. Many things like this one are left not precisely defined to follow the text without ambiguity.

p6: Spatial network distance: if I understand correctly, the spatial network distance is defined by the a number of edge in the cell network of the embryo. If this is correct, I strongly suggest not to use the term spatial, that usually refers to some euclidean or at least metric-based embedding space (spatial would mean here that you intend to use cell geometry or coordinates). What you use here is usually called a topological distance (just using the adjacency connections in the network). The whole paragraph spatial network distance is ambiguous in this respect (for instance why are you speaking of cell spatial positions line 198-194 if they are not used ?). Consider renaming this distance (spatial is not adequate), e.g. network topological distance, or embryo cell-to-cell topological distance (to be completely explicit).

p6: The notations "< x_A >" (e.g. line 184, 205) and "< x_i>_{i in N_A}" (line 206, 207), are different: in one notation one can see the set over which the index varies and in the other, one doesn't see. This type of variation in the notation of the same mathematical notion is confusing.

p6: I would strongly advocate that you consider a careful, precise and unambiguous rewriting of this entire section (starting just after the title "Materials and methods" and finishing just before "Calculation of the contributions Wmosaic and Wregulative") and present the ideas in a more streamlined way. I would also advise the authors to have it proof-checked by some expert with mathematical background in this domain.

p7 L241: "dt = 1": what is the unit ? in particular with respect to a cell cycle that you mention several times in this paragraph.

p7 L247: if epsilon(t) is drawn from a normal distribution, it means that the maximum value of epsilon(t) is less than 1. This in turn means that the maximal rate of change of v(t) is less than 1, and therefore that the amplitude of change of v(t) depends only on dt (as one assumes that the variation of epsilon as a multivariate normal distribution(N(0,sigma)) is taking into account the stochasticity only). This is strange, as one would expect that this rate of change is controlled by a parameter (i.e. an acceleration) that would control the size of RW steps. Did the authors used dt in place of this acceleration parameter ? Do they consider that this is not necessary as RW are self-similar processes ? and why ? Please explain the logics that drives these choices.

p 10 L374-376 and L384-387 say exactly the same thing.

Fig2: The term "spatial context" is used, while it is not defined in the text. "Spatial" should probably be removed.

p12 L:466 and L467: your 5 configurations contains twice the same entry "unrelated cells": Does this mean that there are actually 4 interesting configurations and not 5 ? Was the text re-read before resubmitting ?

p12: "Figure 3D shows that this octant comprises between 10 and 30% of cells along": I understood that each point in these octants is a pair of cells. What means 30% of cells ? Do you mean 30% of cell-pairs ?

p12: could the cells in the Red octant precisely be the L/R matching pairs of cells ? (or contain them). If not, where are the pairs of Left-Right cells ?

p13: Remark: for pair of cells in the Red octant (similar contexts), does not this suggest that their neighboring cells may have a high probability to be found also in the Red octant (higher than the pairs of cells in other octants).</x_a>

**Have the authors made all data and (if applicable) computational code underlying the findings in their manuscript fully available?**

Reviewer #1: Yes

Reviewer #4: Yes

PLOS authors have the option to publish the peer review history of their article (what does this mean?). If published, this will include your full peer review and any attached files.

Reviewer #1: No

Reviewer #4: No

**Figure resubmission:**
---

## [Decision Letter · Decision Letter 3]

2 Dec 2025

Dear Dr Villoutreix,

We are pleased to inform you that your manuscript 'Assessing the Relative Contributions of Mosaic and Regulatory Developmental Modes from Single-Cell Trajectories' has been provisionally accepted for publication in PLOS Computational Biology.

Best regards,

Adriana San Miguel

Academic Editor

PLOS Computational Biology

Marc Birtwistle

Section Editor

PLOS Computational Biology

Dear Authors,

Please address the few final minor comments below in your final manuscript check. Thank you for your extensive revision of this manuscript.

Reviewer's Responses to Questions

**Comments to the Authors:**

Reviewer #4: The authors have made an extensive revision of their paper and improved much the clarity and presentation of their method. Altogether, they responded satisfactorily to my main concerns. I recommend the new version of the paper for publication.

Here is a couple of points that I noticed in the revised text:

L 178-179 x_A and x_B vectors and means should be bold.

L 229: i is not defined. Also, if v is interpreted as a speed, one should not add a position and a speed as they do not have the same units. Consider rewriting this sentence as:

"At each step i, the expression vector is updated as follows: x(i + Delta_i) = x(i) + v(i).Delta_i, where Delta_i is defined as a unit of time (Delta_i=1), ... (otherwise the speed is a number with ill-defined units)."

L304: "the contribution the regulative mode" -?-> the contribution of the regulative mode

**Have the authors made all data and (if applicable) computational code underlying the findings in their manuscript fully available?**

Reviewer #4: **No: **I did not see any mention of the availability of scripts/notebooks that they created for the analyses anywhere.

PLOS authors have the option to publish the peer review history of their article (what does this mean?). If published, this will include your full peer review and any attached files.

Reviewer #4: No

---

## [Editor Report · Acceptance letter]

PCOMPBIOL-D-24-01234R3

Assessing the Relative Contributions of Mosaic and Regulatory Developmental Modes from Single-Cell Trajectories

Dear Dr Song,

I am pleased to inform you that your manuscript has been formally accepted for publication in PLOS Computational Biology. Your manuscript is now with our production department and you will be notified of the publication date in due course.

With kind regards,

Anita Estes
